# Origin and Transport Pathway of Dust Storm and Its Contribution to Particulate Air Pollution in Northeast Edge of Taklimakan Desert, China

**Aishajiang Aili [1], Hailiang Xu [1,*], Tursun Kasim [2] and Abudumijiti Abulikemu [1]**

[1] State Key Laboratory of Desert and Oasis Ecology, Xinjiang Institute of Ecology and Geography, Chinese Academy of Science, Urumchi 830011, Xinjiang, China; aishajiang@ms.xjb.ac.cn (A.A.); abdimijit@ms.xjb.ac.cn (A.A.)

[2] College of Resources and Environmental Science, Xinjiang University, Urumqi 830046, Xinjiang, China; turtsun_kasim@aliyun.com

\* Correspondence: xuhl@ms.xjb.ac.cn; Tel.: +86-991-782-7350; Fax: +86-991-788-5320

**Abstract:** The Taklimakan Desert in Northwest China is the major source of dust storms in China. The northeast edge of this desert is a typical arid area which houses a fragile oasis eco-environment. Frequent dust storms cause harmful effects on the oasis ecosystem and negative impacts on agriculture, transportation, and human health. In this study, the major source region, transport pathway, and the potential contribution of dust storms to particulate air pollution were identified by using both trajectory analysis and monitoring data. To assess the source regions of dust storms, 48 h backward trajectories of air masses arriving at the Bugur (Luntai) County, which is located at the northeast edge of Taklimakan Desert, China on the dusty season (spring) and non-dusty month (August, representing non-dusty season) in the period of 1999–2013, were determined using Hybrid Single Particle Lagrangian Integrated Trajectory model version 4 (HYSPLIT 4). The trajectories were categorized by *k*-means clustering into 5 clusters (1a–5a) in the dusty season and 2 clusters (1b and 2b) in the non-dusty season, which show distinct features in terms of the trajectory origins and the entry direction to the site. Daily levels of three air pollutants measured at a station located in Bugur County were analyzed by using Potential Source Contribution Function (PSCF) for each air mass cluster in dusty season. The results showed that TSP is the major pollutant, with an average concentration of 612 μg/m$^3$, as compared to SO$_2$ (23 μg/m$^3$) and NO$_2$ (32 μg/m$^3$) in the dusty season. All pollutants were increased with the dust weather intensity, i.e., from suspended dust to dust storms. High levels of SO$_2$ and NO$_2$ were mostly associated with cluster 1a and cluster 5a which had trajectories passing over the anthropogenic source regions, while high TSP was mainly observed in cluster 4a, which has a longer pathway over the shifting sand desert area. Thus, on strong dust storm days, not only higher TSP but also higher SO$_2$ and NO$_2$ levels were observed as compared to normal days. The results of this study could be useful to forecast the potential occurrence of dust storms based on meteorological data. Research focusing on this dust-storm-prone region will help to understand the possible causes for the changes in the dust storm frequency and intensity, which can provide the basis for mitigation of the negative effects on human health and the environment.

**Keywords:** dust storm; trajectory pattern; source region; air pollution; NOAA/HYSPLIT model

## 1. Introduction

The identification of air pollutant sources and their transport pathways are crucial issues for controlling and managing of air pollution [1,2]. There are various advanced methods and techniques to identify the source areas of pollutants [3–5]. Among them, the HYSPLIT model is the most widely applied tool which is capable of establishing source-receptor relationships over long distances, and has already been widely used to identify sources and transport pathways of air pollutants [6–10]. Using backward trajectory

statistics, the air pollutant sources can be identified by analyzing the pollution information together with trajectory information [11,12].

The Taklimakan Desert, with the total area of 337,000 km$^2$, is the major source region of dust storms in China [13–15]. Dust storms have already become frequent and serious natural disasters which can impact the environment, economic development, and public health in the surrounding areas. During the dust storm period, large amounts of particulate air pollutants transported with the dust storm over long distances increase the air pollutant concentration and decrease air visibility [16–18]. Liu et al. analyzed dust storm frequency and its relationships with wind, precipitation, vegetation, and soil moisture conditions in northern China [14]. Kahar et al. analyzed the temporal and spatial changes of dust storm frequency in the Taklimakan Desert [15]. Quan et al. analyzed the distribution and causes of dust storm in China [17]. However, limited research work has focused on assessing the dust storm events in this area; notably, the contribution of the dust storms to the air pollutant levels has still not been investigated. In the previous research works, the entire desert or whole province was considered as the study area. There is lack of systematic scientific efforts here, and most of the available research works so far have focused mainly on the formation and distribution of the dust storm and its negative impact on the environment. There is a need to investigate in-depth the major source regions and air mass pathways during dusty days in this area. There are still insufficient scientific efforts focusing on the impacts of dust storms in specific hot spot regions. This study therefore was designed to partly fill in this information gap.

In this research, dust storms are classified into three types based on their severity degree: suspended dust, blowing dust, and sandstorm [19,20]. The origins and pathway of the dust storm which affect the selected study site (Bugur County) have been investigated, and the contribution of each source to the measured air pollutant levels has been determined. The Bugur County is located between the Taklimakan Desert and Tianshan Mountain, and is surrounded by the desert from the west, east and south site. It can be assumed that the main dust origins affecting Bugur County are expected to be located in these regions. To confirm this assumption and to further determine the boundaries of the areas, HYSPLIT backward trajectory models were used. A total of 48 h of HYSPLIT backward trajectories were computed for the total of 1830 dusty days in the period of 1999–2013. Detailed statistical methods for the HYSPLIT model are given in Section 2.3 of this paper. The result of this research would be helpful to determine more effective controlling strategies of dust storms in this area.

## 2. Materials and Methods

### 2.1. Overall Characteristics of Study Area

In this study, the Bugur County, a typical oasis in the northeast fringe of the Taklimakan Desert, Xinjiang, China, was selected as a study area. The geographical coordinates of this county are between 83°38′–85°25′ E and 41°05′–42°32′ N. The total area is 14,789 km$^2$ and the total population is 113,000 [15,21] The climate condition of this area belongs to temperate continental arid climate with obvious seasonal changes. The monthly average temperature is highest in June (26.4 °C) and lowest in January (−7.0 °C). Annual precipitation is only 57 mm and mostly occurs in the summer, while annual evaporation is 2800 mm. The annual bright sunshine is about 3000 h, and the frost-free period is 210 days. Based on previous studies, the annual frequency of three types of dust storms was 17.47 times/year for suspended dust, 6.05 times/year for blowing dust, and 1.38 times/year for a strong sand-dust storm [20], and approximately 80% of dust storm weather occurs in spring. Therefore, Bugur County belongs to a region in China where dust storms are frequent [20,21]. Although the ecological environment is fragile and the local people are vulnerable to the impact of the dust storms, this area has become an economic strategic zone and traffic center of Southern Xinjiang because of rich oil resources and the great potential for cotton production [22,23].

### 2.2. Data Sources

(1) Meteorological data. Meteorological data in the period of 1999–2013 (dust storm frequency, temperature, wind speed) used in this study were obtained from Bugur Meteorological Station and the China Meteorological Data Sharing Service System (http://cdc.cma.gov.cn/), as well as from the National Oceanic and Atmospheric Administration, Air Resources Laboratory (NOAA ARL), which are available at http://www.arl.noaa.gov/ready.html.

(2) Air pollution data. Air pollution indexes (APIs) as well as daily average levels of $SO_2$, $NO_2$ and TSP in the period of 1999–2008 were obtained from the Environmental Monitoring Center of Xinjiang Uyghur Autonomous Region (XUAR) and Bugur Environmental Protection Bureau. At present, China mainly adopts two methods to collect air pollutant concentrations: sampling and weighing based on filter membrane and continuous automatic measurement by using PM automatic component analysis device PX-375 and dynamic calibrator APMC-370. Such systematic data of other pollutants, such as fine particulate matter, CO, and ozone were not available.

### 2.3. Statistical Analysis

(1) Major source region and trajectory pathway of dust storms. In this section, 2 seasons were considered, the dusty season (include both dusty days and non-dusty days in spring) and non-dusty season (represented by August month). The dusty days were further classified into three types of dust storm weather. For the dusty season, four months (March, April, May and June) were selected from each year in the period of 1999–2013 because of high dust storm occurrence in these months [20]. In total, 1830 days were selected for the HYSPLIT model analysis. The August month from each year in the period of 1999–2013 was selected to identify the typical trajectory for the non-dusty season. In total, 465 days were analyzed for the HYSPLIT trajectories in this non-dusty season. Previous research indicated that, because of the topographic constraint of Taklimakan Desert and surrounding areas, the dust from this region could affect only the basin itself. Therefore, a short-range transportation period (48 h) of dust storms was considered in this study. Using the HYSPLIT model, the 48 h of backward trajectories were obtained for each of the selected days (total of 2295 days). The trajectory started from at the center of Bugur County (41.8° N, 84.2° E) at 9:00 UTC (Coordinated Universal Time) each day, which was equivalent to 15:00 p.m. local standard time (LST). The arriving height of air masses to the starting point was fixed at 500 m above ground level (AGL) [24] in order to minimize the friction effects of the surface. The "model vertical velocity option" which is included meteorological variables was selected when running the HYSPLIT model. The "Final Analysis" meteorological data archives of the Air Resource Laboratory, National Oceanic and Atmospheric Administration (NOAA), which are available online at (http://ready.arl.noaa.gov/hypub-bin/trajsrcm.pl), were used to run HYSPLIT4 model. To classify the air mass trajectories, the *k*-means clustering technique (SPSS 20) was applied by using meteorological variables including ambient temperature (K), potential temperature (K), daily rainfall (mm/day), mixing layer depth (m), relative humidity (RH, %), solar radiation flux ($W/m^2$) and wind speed (m/s) measured at the arriving location (the Bugur County). The coordinates of air mass backward trajectories include latitudes, longitudes, and altitude of the air masses at 4 points on a trajectory: 12 h, 24 h, 36 h, and 48 h. The goal of the *k*-means clustering is to divide the dataset into a certain number of homogeneous clusters (k). The cluster membership can be determined based on the distance between the data point and the kth centroids.

(2) Air pollutants levels associated with the HYSPLIT trajectory pattern. Due to the unavailability of up-to-date air pollutant monitoring data (i.e., after 2009), the 24 h air pollutants ($SO_2$, $NO_2$ and TSP) concentration data during the period of 1999–2008 were used to analyze the air pollution level associated with the HYSPLIT trajectory pattern in the dusty days (301 days). The daily average concentrations of 3 types of air pollutants in dusty days were examined in relation to the HYSPLIT trajectory pattern, and the meteorological characteristics including humidity and thermal characteristics of each

cluster were compared and examined. To examine the possible contribution of dust storms to air pollution, the potential source contribution function (PSCF), which indicates the probability that an air mass on a high air pollution day originated in a given grid cell, was calculated using Equation (1) [25–28]. Based on the PSCF value of different pollutants, the pollutant distribution map was generated using GIS software. In this method, the days with high air pollution levels in the study area were first screened out from a total of 301 dusty days. The high pollution days were defined as the days which the daily average concentrations of air pollutants ($SO_2$, $NO_2$ and TSP) are higher than the mean value of those pollutant concentrations during the dusty days. The PSCF value for a given grid cell is calculated by counting the trajectory segment endpoints that originated within that grid cell. The movement of an air parcel is described as segment endpoints of coordinates in terms of latitude and longitude. The total number of endpoints that fall in the cell is $n_{ij}$. There is a subset of these endpoints having arrival times at the site corresponding to measured pollutant concentrations higher than a given criterion value, i.e., above the average in this study. This number of endpoints was defined to be $m_{ij}$. The domain for PSCF computation, which covered all the starting points of 48 h back trajectories, accordingly stretched from 36° N to 43° N latitude and 76° E to 92° E longitude, containing 118 grid cells of 1° × 1°. The PSCF value for the ijth cell was then defined as:

$$PSCF_{ij} = m_{ij}/n_{ij} \tag{1}$$

where, $n_{ij}$ is the total cases that air masses originate in the ijth cell during the dusty days in study period. $m_{ij}$ is the number of cases that air masses originate the ijth cell on the high pollution days. $PSCF_{ij}$ is the probability that air masses originating in the ijth cell on the high pollution days.

In the PSCF analysis, small values of $n_{ij}$ can produce high PSCF values with high uncertainties. In order to minimize this artifact, an empirical weight function $W_{ij}$ originally proposed by Zeng and Hopke (1989) [29] was applied when the number of the endpoints per a particular cell was less than about three times the average values of the endpoints per cell:

$$W_{ij} = \begin{cases} 1.0 & 4 < n_{ij} \\ 0.7 & 3 < n_{ij} \leq 4 \\ 0.4 & 2 < n_{ij} \leq 3 \\ 0.2 & n_{ij} \leq 2 \end{cases} \tag{2}$$

## 3. Results and Discussions

### 3.1. Occurrence of Dust Storm in Dusty Season

According to the criteria given in AQSIQ and NSC (2006), the 15 years of dust storm record in the period of 1999–2013 for the dusty season (from March to June on each year) were classified into 3 types depending on the severity: suspended dust, blowing dust, and sandstorm. The least-severe type is called suspended dust weather and refers to the suspending of dust in the air under calm or low wind conditions with the atmospheric visibility below 10 km. The more severe dust weather is called blowing dust weather and refers to lower horizontal visibility, 1 km–10 km, while the most severe/strong dust weather is called a sandstorm, which refers to such phenomena when the instantaneous wind velocity is over 25 m/s and horizontal visibility of air is below 1 km. During this period, a total of 490 dusty days were recorded; among them, sandstorm weather occurred on 32 days, blowing dust weather occurred on 116 days, and the suspended dust weather occurred most frequently, on 342 days, out of 1830 days considered. The monthly frequency was the highest in April for all types of dust weather, followed by March, while the lowest frequency was in June among the 4 months of the dusty season.

### 3.2. Patterns of HYSPLIT Trajectories of Air Masses Arriving at the Study Area in Dusty Season

The HYSPLIT backward trajectories were obtained for every day of 1830 days of the dusty season of 15 years (1999–2013). The 48 h backward trajectories were obtained which show the origin and pathways of the air masses arriving at the Bugur County at 9:00 UTC.

Based on the score matrix of these trajectory data, the *k*-means clustering technique has classified the air mass trajectories arriving at the Bugur County on 1830 days of the dusty season of 15 years into 5 clusters that are named in this study by their origins and the entry direction to the study site. The backward trajectories of each of the 5 clusters are presented in Figure 1 showing the ranges of horizontal coordinates representing each cluster.

As seen from Figure 1, the resulting 5 clusters are distinctively different in terms of length, shape, vertical position, and the origins of the 48 h back trajectories. Within each cluster, there is a satisfactory uniformity in the shape, pathway, trajectory length and origins observed. It worth mentioning that the 5 clusters produced by the *k*-means technique match satisfactorily with the visual classification made for the trajectories. The variations of 48 h back trajectories obtained for a particular month are quite typical and do not change much from year to year during the 15 years of the study period. The monthly frequency of three types of dusty days (suspended dust, blowing dust and sand storm) within each cluster showed a consistent trend and are ranked in the following order: April > March > May > June. The occurrence of the clusters (number of days when a cluster was identified) and the average meteorological conditions at Bugur County associated with each cluster are presented in Table 1.

**Table 1.** The total number of days and average meteorological conditions at Bugur County associated with different HYSPLIT clusters in dusty season (March–June of 1999–2013).

| | Parameters | Cluster 1a | Cluster 2a | Cluster 3a | Cluster 4a | Cluster 5a | Unclassified |
|---|---|---|---|---|---|---|---|
| | **Total number of days** | 432 | 127 | 242 | 521 | 282 | 226 |
| | Number of dusty days | 136 | 47 | 43 | 143 | 58 | 72 |
| | Percentage of dusty days (%) | 31.5% | 37% | 17.8% | 27.4% | 20.6% | 31.5% |
| | *Suspended dust* | *86* | *26* | *30* | *106* | *53* | *37* |
| | March | 28 (32.5%) | 8 (30.1%) | 9 (30%) | 27 (25.5%) | 17 (32.1%) | 11 (29.7%) |
| | April | 34 (39.5%) | 14 (53.8%) | 14 (46.7%) | 33 (31.1%) | 27 (50.1%) | 19 (51.3%) |
| | May | 15 (17.4%) | 3 (11.5%) | 5 (16.6%) | 28 (26.4%) | 6 (11.35) | 5 (14.3%) |
| | June | 9 (10.4%) | 1 (3.8%0 | 3 (10%) | 16 (15.1%) | 35.7%) | 2 (5.4%) |
| | *Blowing dust* | *38* | *15* | *10* | *31* | *13* | 33 |
| **Dust storm frequency** | March | 13 (34.2%) | 5 (33.3%) | 3 (30%) | 7 (22.6%) | 4 (30.75) | 9 (27.3%) |
| | April | 15 (39.4%) | 8 (53.3%) | 5 (50%) | 11 (35. %) | 5 (38.5%) | 20 (60.6%) |
| | May | 7 (18.4%) | 2 (13.3%) | 1 (10%) | 9 (29%) | 3 (23.1%) | 4 (12.1%) |
| | June | 3 (7.9%) | 0 | 1 (10%) | 4 (12.9%) | 1 (7.7%) | 0 |
| | *Sand storm* | *12* | *6* | *3* | *6* | *3* | *2* |
| | March | 3 (25%) | 2 (33.3%) | 1 (33.3%) | 1 (16.7%) | 1 (33.3%) | 0 |
| | April | 7 (46.7%) | 4 (66.7%) | 2 (67.7%) | 2 (33.3%) | 2 (66.7%) | 2 (100%) |
| | May | 2 (16.7%) | 0 | 0 | 2 (33.3%) | 0 | 0 |
| | June | 0 | 0 | 0 | 1 (16.7%) | 0 | 0 |
| | *Non-dusty days* | 296(68.5%) | 80(63%) | 199(82%) | 378(72.6%) | 224 (79%) | 154(69%) |
| | Potential temperature (K) | 305 ± 22.4 | 298 ± 23.3 | 295 ± 31.1 | 308 ± 19.6 | 307 ± 20.1 | 291 ± 17.8 |
| | Ambient temperature (K) | 283 ± 20.1 | 278 ± 16.2 | 267 ± 14.3 | 290 ± 24.8 | 281 ± 24.2 | 278 ± 17.5 |
| *Meteorological condition* | Rainfall (mm/day) | 0 | 0 | 0.48 | 0 | 0.24 | 0.24 |
| | Mixing layer depth (m) | 1254 ± 221 | 1224 ± 103 | 1090 ± 95 | 2121 ± 286 | 1688 ± 148 | 1312 ± 87 |
| | Relative humidity (%) | 23 ± 6.3 | 24 ± 4.8 | 43 ± 7.7 | 31 ± 7.5 | 23 ± 5.6 | 26 ± 6.8 |
| | Downward solar radiation flux (W/m$^2$) | 1216 ± 67 | 1526 ± 122 | 743 ± 57 | 713 ± 86 | 664 ± 49 | 1683 ± 146 |
| | Wind speed m/s | 8.6 ± 1.2 | 7.9 ± 0.6 | 5.8 ± 0.6 | 5.4 ± 0.7 | 6.4 ± 1.1 | 7.3 ± 0.9 |

Note: denotion "a" of the cluster name, e.g., cluster 1a, is used to differentiate with those in non-dusty season which carry a denotion "b".

Table 1 also shows the frequency of dust storms of different intensity (3 types of dust weather) during the dusty season. It is noted that the clusters are characterized by different meteorological conditions measured at Bugur County as seen in Table 1. The monthly occurrences of the clusters are also different. All other clusters occur more frequently in April. A discussion of the origins, transport pathways, and meteorological characteristics measured at Bugur County for each cluster is given below.

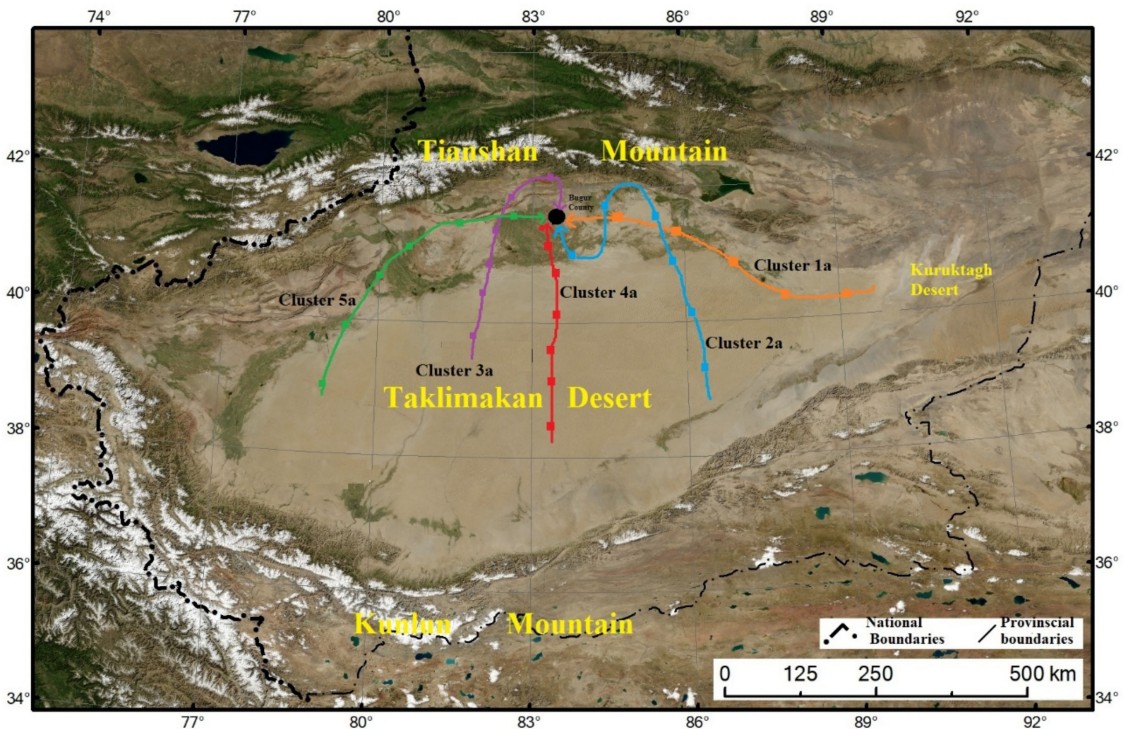

**Figure 1.** The origins and pathways of 48 h backward trajectory for different clusters in the dusty season.

(1) *Cluster 1a: E-E category*

Air masses of this category normally originate from the east, e.g., the Turpan Basin, at an average altitude of around 1100 m, which is the lowest of all air mass clusters (Figure 1), and move eastward through Kuruktagh Desert, then over Lopnur county, Korla city, and finally arrive at Bugur from the east. This cluster occurs with the second-highest frequency of all clusters; it was observed on 432 days out of 1830 total days of the dusty season of the 15 years (around 23.6%), and has a relatively high occurrence frequency in both April and March (accounts for 72% of days). It has a long pathway above the desert areas and a short pathway over the populated areas. It can be characterized by the dry and hot air masses with the high ambient temperature (283 K) and no rainfall (0 mm/day) and low relative humidity (23%) (Table 1). The mixing height (1254 m) at Bugur shows a lower value, which may be related to the pressure field.

(2) *Cluster 2a: SE-S category*

This air mass mostly starts from the east edge of the Taklimakan Desert with higher altitude (the average altitude of >2000 m at the starting point, Figure 1) and moves northward, reaching the north part of the Kuruktagh Desert, then turns southwest through the Kuruktagh Desert and moves westward along the Taklimakan Desert and Lopnur County, finally arriving at Bugur from the south. This category occurs with the lowest frequency (127 days or 6.9% of the examined days) of all clusters. This cluster occurs most frequently in April. The characteristics of air masses in this cluster are somewhat similar to those in cluster 1a, but with lower ambient temperature (278.4 K), almost the same relative humidity

(24%), and also without rain (0 mm/day) (Table 2). The solar radiation flux (1526 W/m$^2$) was the highest in all clusters while wind speed (7.9 m/s) was the second-highest.

(3)   *Cluster 3a: S-N category*

Air masses in this cluster originate from the middle part of the Taklimakan Desert with higher altitude (>1000 m) and move northward to the Tangri mountain area, turn east, and then arrive at Bugur from the north direction (Figure 1). This cluster occurs with the second-lowest frequency of all clusters (242 days or 13.2%). This air mass type can be characterized by a cool and moist air mass with the lowest ambient temperature of 267.5 K, highest rainfall (0.48 mm/day), and highest relative humidity (43%) among all clusters (Table 1). The wind speed is moderate and the mixing height is the lowest (~1100 m).

(4)   *Cluster 4a: S-S category*

The origins of trajectories in this cluster are approximately similar with cluster 3a, but with a longer pathway over the desert areas. The air mass within this cluster originates from the middle part of Taklimakan Desert and moves northward and arrives at Bugur from the south (Figure 1). These types of air masses were observed on 521 days with the highest frequency (accounts for 28.5% of total days). With its long pathways above the desert areas, these air masses have the potential to bring large amounts of sand and dust particles to Bugur County from the south and southwest directions. This air mass type is hot and dry with the highest ambient temperature (290.4 K), no rainfall, and moderate relative humidity (31%). The mixing height is the highest of all clusters (>2000 m), while the wind speed was the lowest (5.4 m/s).

(5)   *Cluster 5a: SW-W category*

Air masses within this cluster normally originate from west and southwest part of Taklimakan Desert at the highest altitude (>5000 m), move northward passing over the west edge of the Taklimakan Desert, then turn northeast, passing over the populated areas such as Aksu city, Toksu county, and Kucha county before arriving at Bugur County from the west (Figure 1). This type was observed on 282 days out of 1830 total days (accounts for 15.4%). This cluster can be characterized by more moist air masses with a relatively higher rainfall (0.24 mm/day) and but low relative humidity (23%), second highest mixing height (1688 m), and moderate wind speed (6.4 m/s).

The number of days that cannot be classified (unclassified) in this study is still considerable (226 days or 12.5%). Thus, a future study should improve the classification scheme by, for example, using more surface and upper air meteorological stations to better capture the local and regional features.

In summary, the air masses are mainly coming from the east and south to the study area in the dusty season. Air masses arriving at the study area from the east (cluster 1a) and south (cluster 2a and 4a) accounted for 65% of the total number of the considered trajectories. Comparing the meteorological characteristics of the 5 clusters, there are clear differences in the moisture conditions and ambient temperatures. For example, cluster 3a, which comes from the north, is characterized by cool and wet air masses with the lowest temperature and highest rainfall and relative humidity; this is mainly associated with its pathway over the cool and moist mountain area in the north. On the opposite, cluster 2a and 4a, which come from the south, were characterized by hot and dry air masses with higher temperature and lower relative humidity and rainfall. This is also mainly associated with their longer pathway over the desert area with hot and dry meteorological conditions. As for the percentages of dusty days in each cluster, cluster 3a shows the lowest dust storm frequency (17.8%) among the 5 clusters. This could be related to its pathway over the cool mountain area with higher vegetation cover before arriving at the study site. The dust storm frequency was the highest in cluster 2a (>37%), followed by cluster 1a, cluster 4a, and cluster 5a. The air masses of these two clusters have a longer pathway above the desert area, and hence, may carry sand-dust particles from the desert area to the study site. Both clusters 1a and 2a also pass over some populated areas; hence, they may also bring in nearby pollution to the study area.

Comparing the monthly frequency of dust storms in each cluster, there are no significant differences; about 70% of dust storms occur in April and March in each cluster.

### 3.3. Patterns of Air Mass HYSPLIT Trajectories in Non-Dusty Season

To identify the air mass trajectories arriving at Bugur County in non-dusty season, the air mass 48 h backward trajectories during the non-dusty season, determined for 465 days (August month of each year from 1999 to 2013), were used to compare with the backward trajectories during the dusty season. The detailed information about the total number of observed days within each cluster and their meteorological characteristics during this period are shown in Table 2.

**Table 2.** The total number of days and meteorological conditions of each cluster in the non-dusty season (August, 1999–2013).

| Parameters | Cluster 1b | Cluster 2b | Unclassified |
|---|---|---|---|
| Total number of days | 243 | 135 | 87 |
| Potential temperature (K) | 304 ± 31.4 | 344 ± 33.6 | 291 ± 24.5 |
| Ambient temperature (K) | 283 ± 22.6 | 311 ± 28.8 | 278 ± 23.0 |
| Rainfall (mm/day) | 1.92 ± 0.2 | 0.87 ± 0.1 | 0.47 ± 0.1 |
| Mixing layer depth (m) | 1054 ± 86.1 | 2688 ± 179.3 | 812 ± 66.5 |
| Relative humidity (%) | 48 ± 3.6 | 30 ± 2.4 | 34 ± 2.8 |
| Downward solar radiation flux (W/m$^2$) | 629 ± 51.2 | 664 ± 52.6 | 683 ± 60.4 |
| Wind speed (m/s) | 4.4 ± 0.5 | 5.3 ± 0.6 | 4.6 ± 0.4 |

The PCA with *k*-means clustering technique was applied to classify the air mass trajectories arriving at the Bugur County on these 465 days into 2 clusters which were distinctively different in terms of length, shape, vertical positions, and the origins of the back trajectories (Figure 2). These clusters are named cluster 1b and cluster 2b to distinguish them from those of the dusty season presented above. The origins, transport pathway, and meteorological characteristics of each cluster measured at Bugur County are described as follows.

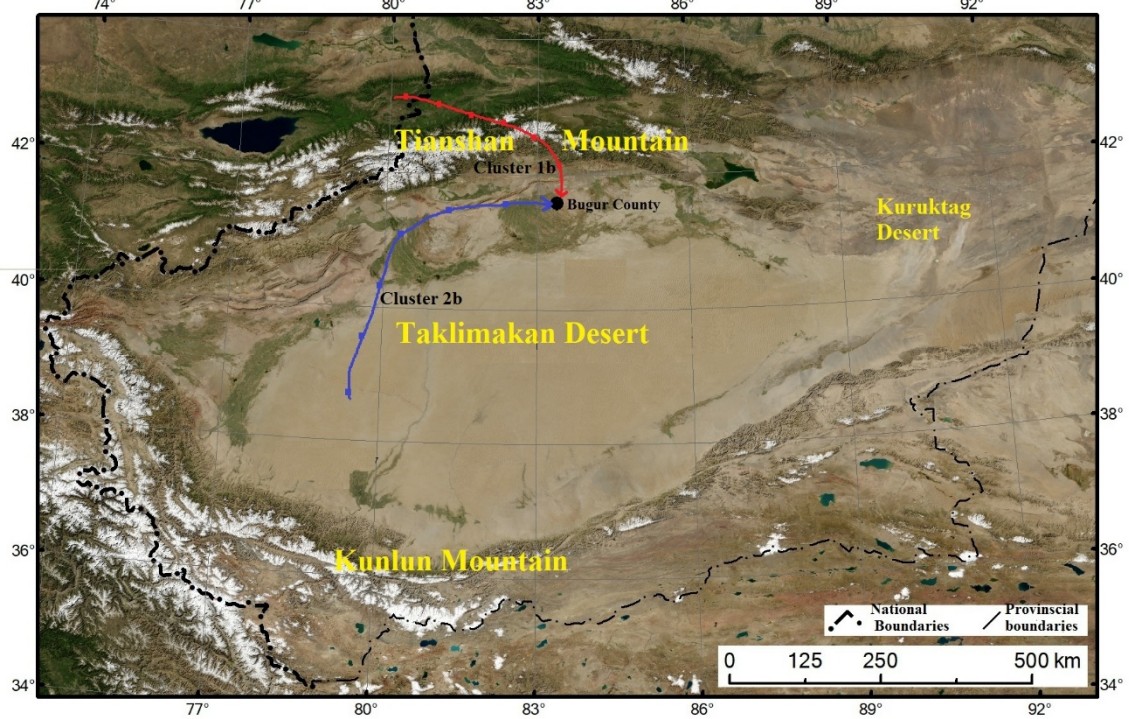

**Figure 2.** Origins and pathways of 48 h backward trajectory for different clusters in the non-dusty season.

(1)    *Cluster 1b: NW-N category*

Air masses in this cluster originate from the valley area between Kazakhstan and western Xinjiang at high altitudes (2100 m–2200 m) and move southeast, passing over the Tangri Mountain at high altitudes and sinking down when turning southward, finally arriving at Bugur from the north (Figure 2). This cluster was observed on 243 days (it accounts for 52.2% of the total). It has a long pathway above the mountain areas. It can be characterized by moist and cool air masses with higher rainfall (1.92 mm/day), higher relative humidity (48%), and lower ambient temperature (283 K) than cluster 2b (Table 2).

(2)    *Cluster 2b: SW-W category*

The origin and pathway of air mass trajectories in this cluster are quite similar with cluster 5a in the dusty season. Air masses within this cluster normally originate from the southwest part of the Taklimakan Desert at lower altitudes (550 m–650 m), move northward passing over the west part of the Taklimakan Desert, then turn to the east and move along the south slope of the Tangri Mountain and arrive at Bugur County from the west (Figure 2). This type of cluster was observed on 135 days out of 465 total days (it accounts for 29%). As compared to cluster 1b, this cluster can be characterized by a relatively lower rainfall (0.87 mm/day), lower relative humidity (30%), and higher ambient temperature (311.6 K) and higher mixing height observed at the Bugur county (Table 2).

*3.4. Comparison of Air Mass Trajectories between Dusty and Non-Dusty Season*

(1)    *Origins and pathways of trajectories*

During the dusty season, backward trajectories indicate that the air masses mostly originate from the east and south part of the Taklimakan Desert, and a majority (around 85%) arrive at the study area from the east and south. These air masses have the potential to bring large amounts of sand and dust particles to the study area from the desert. During the non-dusty season (represented by August), the air masses mostly come from the north, i.e., cluster 1b, with 52% occurrence. In this cluster, the air masses originated from the valley area in the north and pass over the mountain before arriving at the study area. Although the air masses of cluster 2b in the non-dusty season have a similar trajectory pathway and origin as those of cluster 5a of the dusty season, they are expected to be different in terms of the associated air pollution levels. This is because, in the non-dusty season, most of the areas passed over by the air masses are covered by either cultivated or natural vegetation cover; hence, they do not carry sand-dust particles to the study site.

(2)    *Moisture condition*

During the non-dusty season, the relative humidity and rainfall of air masses are much higher than in the dusty season because the season normally has higher rainfall. Similar research has been conducted in the Taklimakan Desert area and also confirmed this result. Pan et al. analyzed the relative humidity of air during the dusty season and found that the average air humidity is lower in sandstorm days (0.0386–0.6306) than blowing dust and suspended dust (0.1079–1.00) [30]. It can be seen from Tables 1 and 2 that there are remarkable differences between the meteorological conditions of air masses in the dusty season and non-dusty season. The average relative humidity and rainfall show higher values in the non-dusty season, i.e., for August, 39% and 1.79 mm/day, respectively, as compared to the dusty season values of 26.2% and 0.16 mm/day, respectively (Tables 1 and 2). In principle, higher rainfall and relative humidity in the non-dusty season enhance vegetation growth and can increase soil moisture and reduce the likelihood of dust events.

(3)    *Wind speed*

Wind is the major dynamic force of dust storm formation. There is a difference in average wind speed between the dusty season ($6.9 \pm 0.5$ m/s) and non-dusty season ($4.8 \pm 0.5$ m/s) (Tables 1 and 2). During the dusty season, higher wind speed results from either atmospheric circulation or the local temperature gradient [31], which plays an important role in dust storm formation in this area.

### 3.5. Air Pollutants Concentration Associated with Different Clusters in Dusty and Non-Dusty Seasons

Due to the unavailability of more recent air pollutant monitoring data (i.e., after 2009), the 24 h air pollutant concentration data during the period of 1999–2008 were used to analyze the air pollution levels associated with each air mass cluster. The daily average concentration of $SO_2$, $NO_2$, and TSP measured at Bugur County during the dusty season (March–June) and non-dusty season (August) associated with each cluster are presented in Figure 3. In addition, this study also calculated the average levels of non-dusty days of the dusty season for comparison.

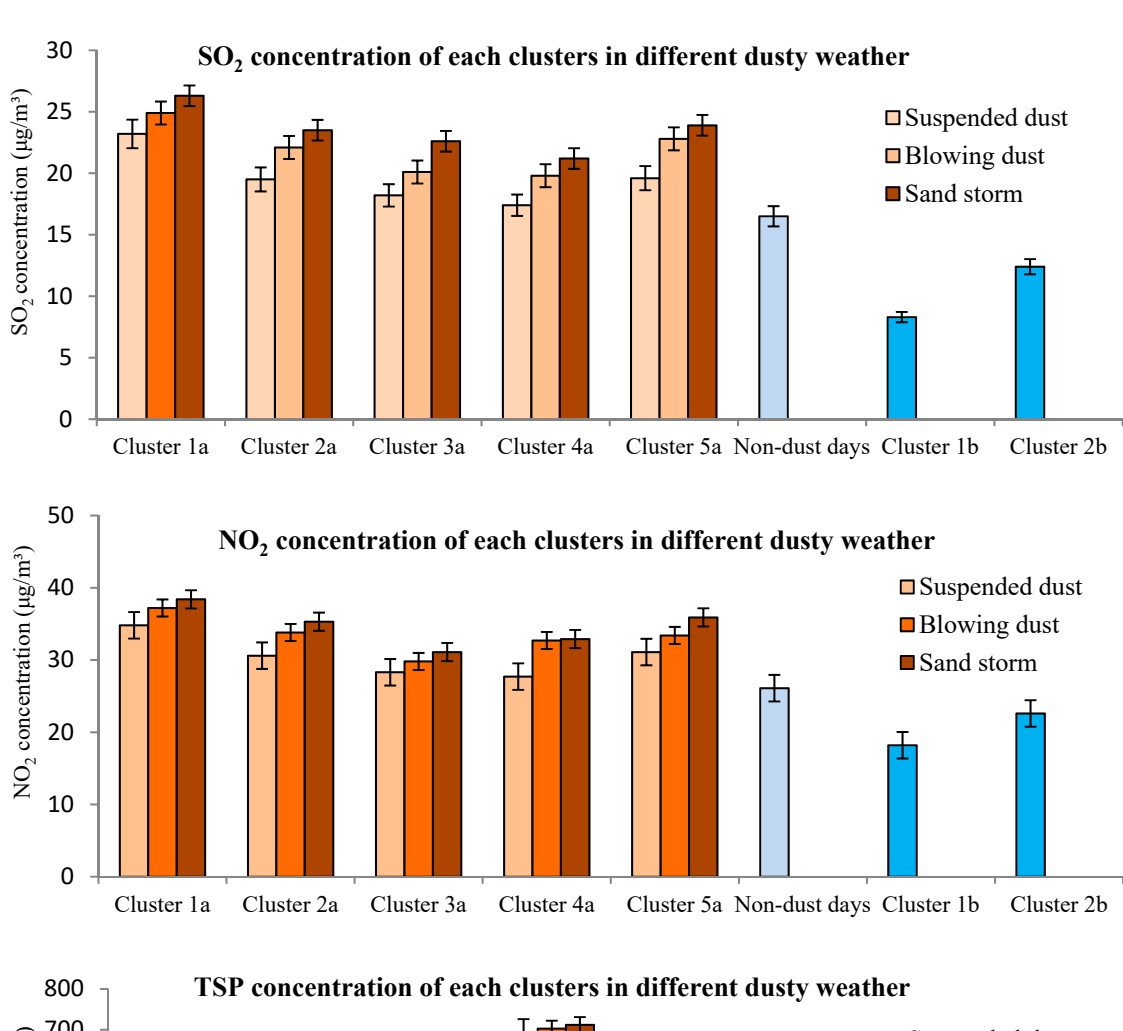

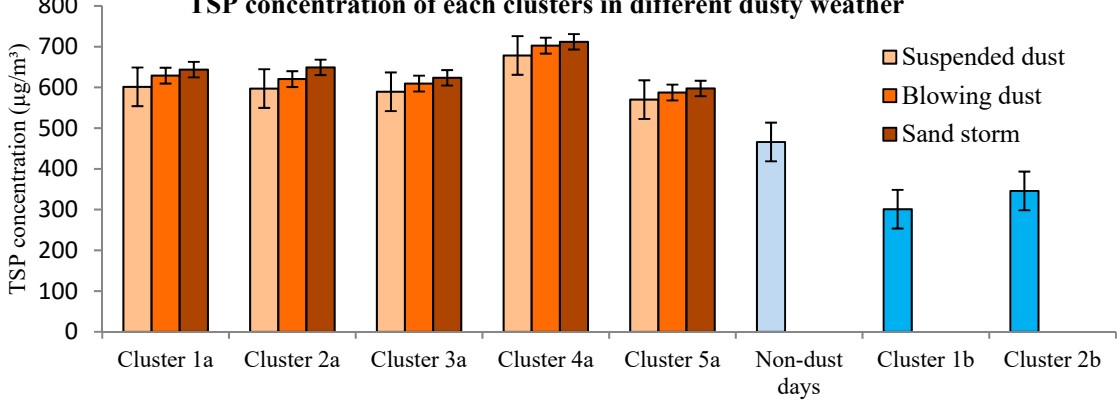

**Figure 3.** Daily average concentrations of 3 types of air pollutants associated with different clusters of the dusty season (cluster a) and non-dusty season (cluster b). Note: Non-dust days are normal days (no dust weather reported) during the dusty season (March–June).

It can be seen from Figure 3 that in the dusty season, all pollutants were increased with the dust weather intensity, i.e., from suspended dust to dust storm. This result is consistent with previous research. For example, Yu et al. indicated that the air pollutants in southern Xinjiang presented distinct seasonal and spatial distribution characteristics. The highest PM concentrations were found in the period of dust storm days and the lowest PM concentrations were found in the period of non-dusty days [32]. The average levels observed during non-dusty days (belonging to different clusters) were the lowest. The average levels of all pollutants during the non-dusty season were lower than in the dusty season. The lowest levels were observed for the cluster 1b.

$SO_2$ concentrations were lower between regular days (19 $\mu g/m^3$) as compared to dusty days (average for all clusters, 22 $\mu g/m^3$). Comparing the $SO_2$ concentration between each cluster, the highest $SO_2$ concentrations were observed for cluster 1a, followed by cluster 5a, then by cluster 2a and 3a, while the lowest $SO_2$ concentrations were observed in cluster 4a, which can be explained using the air mass trajectories given above. For example, cluster 1a (comes from the east direction) and cluster 5a (comes from west direction), associated with higher $SO_2$ concentrations, commonly had a longer pathway over the populated areas, including Lopnur County, Korla City, and Karashahar (Yanji) County in the east, and Aksu City and Kucha County in the west, where the industrial activities are concentrated. The lowest $SO_2$ concentrations were associated with air masses from the southeast (clusters 2a), north (cluster 3a), and south (cluster 4a), which enter Bugur County without passing the populated areas. Consistent with this analysis, the same situation was observed for the $NO_2$ concentrations in each cluster (Figure 3). For example, the air masses of cluster 1a and cluster 5a which had a longer pathway over the anthropogenic pollution areas showed the highest (37 $\mu g/m^3$) and second-highest (34 $\mu g/m^3$) $NO_2$ concentrations. The air masses of cluster 2a, cluster 3a, and cluster 4a, which had a shorter pathway over or without passing over the anthropogenic pollution areas, showed lower $NO_2$ concentrations. Actually, E-E (cluster 1a), S-S (cluster 4a), and SE-S (cluster 2a) are three categories of air masses originating from more arid areas with dry weather conditions. The trajectory pathways can enhance the entrance (pick-up) of anthropogenic air pollutants by air masses and transport the pollutants to the study site. In contrast, if the air masses pass over unpolluted areas, especially over the mountains or valley areas with vegetation coverage, fewer pollutants could be picked up and transported by air masses to the study site. Previous research also confirmed the contribution of dust storms to the air pollution level. Flores et al. studied the influence of desert dust transport on surface $PM_{10}$ concentrations in Istanbul, Turkey, and found that averaged-daily $PM_{10}$ concentrations due to desert dust transport ranged from 3 to 126 $\mu g/m^3$ during the 2007–2014 study period [31]. Hesam et al. indicated that topography and land use have a significant role in synoptical pattern changes and strongly influence air pollutant transportation in the atmosphere by changing the direction and speed of the airflows [11]. It was expected that, during the dust storms, the gaseous pollutant levels including $SO_2$ and $NO_2$, which are mainly formed from man-made combustion activities, particularly in vehicle engines and industry, would not be increased. However, the levels of these pollutants were also higher during dusty weather, which is explained by the air mass trajectory pathways. As expected, TSP concentration (four-month average) shows the highest value (695 $\mu g/m^3$) in cluster 4a, which had a longer pathway over the desert areas (Figure 3). The air masses of cluster 4a originate from the south part of the Taklimakan Desert and move northward, passing over the whole middle of desert and directly arriving at Bugur County from the southwest. The middle part of this desert is composed of shifting sands which are easily blown by the wind into the air. Therefore, cluster 4a is associated with a shifting sand desert, hence including high TSP to the study area.

### 3.6. Distant and Regional Sources of Air Pollutants

The average concentrations of 3 types of air pollutants during the four months of the dusty season (1999–2008) are: $21 \pm 9$ $\mu g/m^3$ for $SO_2$, $33 \pm 2.8$ $\mu g/m^3$ for $NO_2$ and $622 \pm 76$ $\mu g/m^3$ for TSP. In total, 163 days with $SO_2$ levels above the average concentrations

for the study period, 152 days for $NO_2$, and 144 days for TSP were sorted out and are classified as high-pollutant days. The distribution of the high-pollutant ($SO_2$, $NO_2$ and TSP) days in different clusters was also compared. The majority of high $SO_2$ and $NO_2$ concentration days belonged to cluster 1a (46 days for $SO_2$ and 38 days for $NO_2$) and cluster 5a (40 days for $SO_2$ and 33 days for $NO_2$). The majority of high TSP days were observed in cluster 4a (51 days), which agreed with the discussion of air mass trajectories given above. The PSCF (0–1) results are displayed in the form of maps of the area of interest using a color scale. The potential source regions of $SO_2$, $NO_2$, and TSP are shown in Figure 4, respectively.

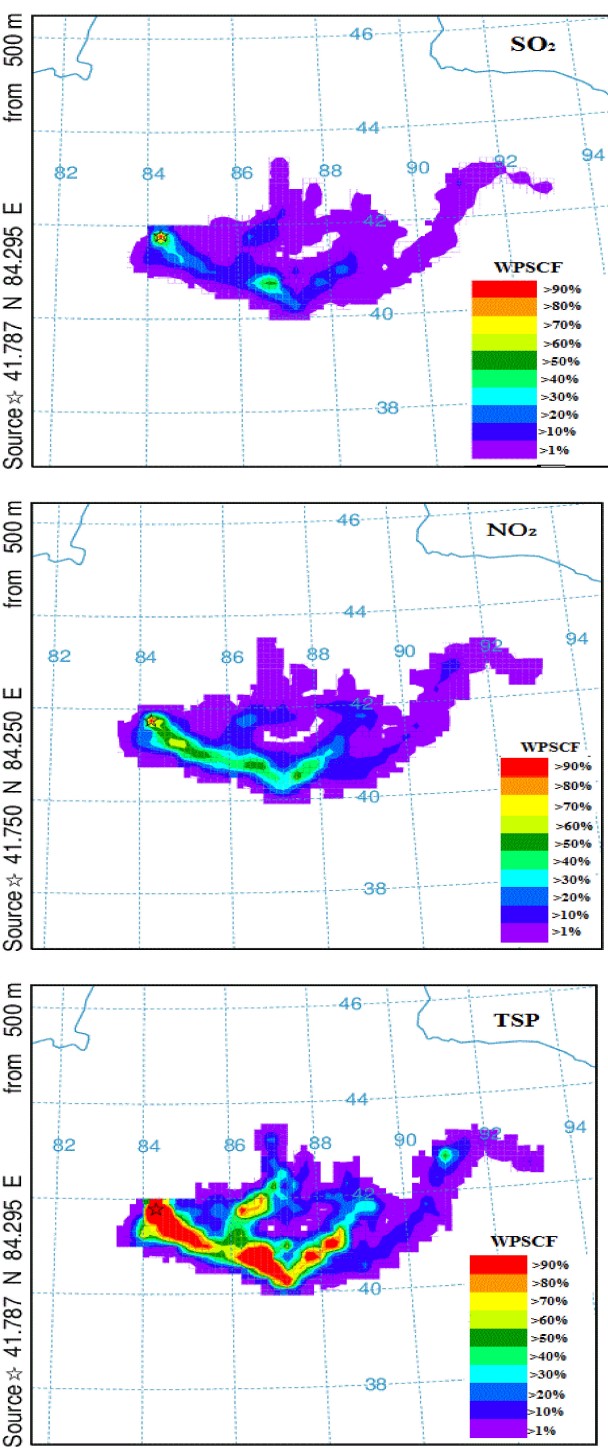

**Figure 4.** The potential source contribution function plot for different pollutants. Note: The asterisk "☆" in the figure is the study area (Bugur County).

The higher probability (higher $PSCF_{ij}$) for regions of high $SO_2$, i.e., the $SO_2$ potential source region, was observed over the north edge and middle part of Kumtagh desert, the west part of the Taklimakan Desert, and nearby populated areas. A similar pattern was also observed for the potential source region of $NO_2$. Thus, the majority of potential source regions for $SO_2$ and $NO_2$ are located in the west and east regions where human development activities are intensive. In comparison, the PSCF map for TSP (Figure 4) shows a different pattern than $SO_2$ and $NO_2$. TSP source regions have less association with anthropogenic polluted areas. Most of the TSP potential source regions are located at the south and middle part of the Taklimakan Desert, where the shifting sands dominate. Thus, the air masses originating in or passing these areas are likely to pick up the pollutants and transport them to Bugur County.

Cluster 4a, on the other side, brings in air masses through some rural areas south of Bugur, and also has a longer pathway over the desert area (distant sources), which may explain the highest levels of TSP in this cluster. Cluster 5a comes from the west to the study site and passes some oil refinery and coal mine sites located in the west part of Bugur. This cluster also has a longer pathway along the highway but short pathway over the desert which may explain somewhat high levels of $NO_2$ and $SO_2$ but low TSP associated with this cluster. Note that cluster 2b has a similar pattern with cluster 5a but has lower levels of all pollutants due to the rainy conditions of August and green coverage of the soil. Cluster 1b has even lower pollution levels than 2b, which may be explained by fewer local contributions, as it comes from the north. It is noted that Cluster 3a also has a large part of the trajectory over the northern mountain area but originated from the desert area in the south. It has moderate pollution levels compared to other clusters in the dusty season.

## 4. Conclusions

(1) *k*-means clustering of air masses classified the backward trajectories of air masses arriving at Bugur County in the dusty season in five patterns. Among them, cluster 2a, 1a, and 4a mainly originated from desert areas, i.e., the main potential source regions of dust storms, hence having higher dust storm frequency. Cluster 3a and 5a, which also originated from the desert area but with longer pathways over the non-desert area before entering the Bugur County, have lower dust storm frequency (below 20%). The air masses arriving at Bugur County in non-dust month of August were classified into two categories. Among them, the cluster 1b category showed more humid and cool air than cluster 2b.

(2) TSP was the main pollutant in the study area, indicating that air pollution was mainly caused by dust. Higher concentrations of $SO_2$ and $NO_2$ were associated with air masses originating from the east and passing through the populated areas (cluster 1a), which shows the potential contributions of anthropogenic sources both local and distant to the site. In contrast, the higher concentration of TSP was associated with air masses originating from the desert area in the south (cluster 4a) passing over the shifting sand dominated desert areas, which pick up and transport the pollutant to the study area.

(3) In the non-dusty month of August, the air pollutant levels were remarkably lower. Air masses in the non-dusty season had high relative humidity and lower wind speed. Moreover, most of the area under the pathway of the air masses was largely covered by either cultivated or natural vegetation, limiting the emission of sand-dust particles carried to the study site.

(4) The results of this study will help to understand the possible causes for the changes in the dust storm frequency and intensity, which can provide the basis for the mitigation of their negative effects on human health and the environment. Our results agree qualitatively or quantitatively with other studies. Based on the dust storm source regions and pathways identified in this study, there should be a focus on increasing the vegetation cover, such as by afforestation and reforestation to partly interrupt and mitigate the dust storm effects and protect the study area. In particular, an increase of the forest cover on the eastern and southern part of the study area which is located on entrance pathways of the air masses in the dusty season is recommended. Appropriate measures/policies should be developed to

protect the ecologically fragile zone between the oasis and desert, which should be linked to long-term planning for development such as population growth, industrialization, shelterbelt establishment, agricultural planning including water-saving agriculture, and so on. To increase the accuracy of trajectory analysis, more variables such as the moisture condition of the surface, wind direction, etc. at the study area can be included for cluster analysis. The southern edge of the Taklimakan Desert e.g., Hetian oasis and Minqin City of Gansu province, China, will be more suitable for further research due to more serious dust storm events and lack of related research in these areas. Satellite data can be used to characterize air pollution levels and also to reveal the long-range transport sources affecting the study area. Systematic ground-based air pollution data, including CO, NO, $O_3$, $PM_{10}$ and $PM_{2.5}$ for a longer period, should be considered to further research the links between the dust storms with air pollution.

**Author Contributions:** Conceptualization, A.A. (Aishajiang Aili); Data curation, H.X.; Formal analysis, T.K. and A.A. (Aishajiang Aili); Methodology, H.X.; Project administration, A.A. (Abudumijiti Abulikemu) and H.X. All authors have read and agreed to the published version of the manuscript.

**Funding:** This research was financially supported by the Western Light Foundation of Chinese Academy of Sciences (Y734341).

**Institutional Review Board Statement:** Not applicable.

**Informed Consent Statement:** Not applicable.

**Data Availability Statement:** The data that support the findings of this study are available from the corresponding author, upon reasonable request.

**Conflicts of Interest:** The authors declare no conflict of interest.

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
