# Peer review of "Origin and Transport Pathway of Dust Storm and Its Contribution to Particulate Air Pollution in Northeast Edge of Taklimakan Desert, China"

_atmosphere, doi:10.3390/atmos12010113_

Round 1

Reviewer 1 Report

This study analysed the dust storm origin and it’s impact on air pollution in the northwest part of China. The overall pollution was investigated by using trajectory and overall pollution data. The study is important from air pollution point of view and the proper analysis of the dust storm source could potentially improve the knowledge of the field.

  1. Authors provided sufficient information in the abstract section. However, the key message is missing in the abstract. Authors need to highlight why this study is important and how it will improve the knowledge of the field?
  2. Authors discussed some literature and the story behind why this study is needed is not clear yet. Authors need to perform more comprehensive literature review and should improve the introduction section. More relevant literature needs to include in the introduction section.
  3. Authors mentioned, they obtained the meteorological data for the period 1999-2013 and air quality data from 1999-2008. How will they compare these data? Why authors are not using the recent data as the pollution significantly increased these days.
  4. In section 3.2, the authors mentioned that the 48 hours back trajectories is quite similar for a wide range of period. It could be similar. However, the authors can explain this issue a bit more for the readers.
  5. There are 3 figures in figure 3 and it’s hard to correlate the discussion as there is no sub-caption for the figures.
  6. Authors mentioned, ‘The average relative humidity and rainfall show higher values in non-dusty season, i.e., for August month: 39% and 1.79 mm/day, respectively, as compared to the dusty season values of 2% and 0.16 mm/day, respectively’. Did the authors present any table or figure for this information?
  7. Conclusion section needs to improve. Authors need to write down the future perspective of this study.

Reviewer 2 Report

Authors’ efforts in identifying the air mass transport pathway and their potential contribution to dust storm/ particulate air pollution by using trajectory analysis in the area (the Bugur County) for multiple years is appreciable.

 Although there is no novel thing except the model data analysis for the area, the first time, however, the reported results could be valuable to local air quality problems/mitigation efforts, health risk assessments, and also regional climate change modeling.

Overall, the article is reasonable, well-balanced, and discussions are proper. However, some minor concerns (mentioned below) should be addressed before considering it for the final publication. So I would recommend this paper for its publication subjected to some minor revision.

Minor Comments

My concerns are not about the technicality of the study but about some minor presentation issues:

  • The abstract is more descriptive of methodology, not of the study’s results.
  • The introduction needs a litter more description of the study background, the need/significance of the study, and the relevant literature review with the relevance of this study.
  • The discussion of results is also missing the citation and comparison of this study with relevant literature.
  • Which monitors had been used for measuring the reported pollutant at XUAR center (in section 2.2.2).
  • Which statistical tool has been used to calculating and mapping of PSCF (in section 2.3.2)
  • In section 3.1, how were classified 3 types of dust storms: suspended dust, blowing dust and sand storm, mention here shortly the followed AQSIQ/NSC criteria.
  • In the conclusion section, some implications of the study results would be beneficial to the readers.
  • Also, please revise the text for some grammatical and style errors.

Round 2

Reviewer 1 Report

Authors revised the manuscript based on the reviewer suggestions and it can be accepted for publication.
